# RNA from a simple-tandem repeat is required for sperm maturation and male fertility in *Drosophila melanogaster*

Wilbur Kyle Mills[1,2], Yuh Chwen G Lee[1,2,3], Antje M Kochendoerfer[4], Elaine M Dunleavy[4], Gary H Karpen[1]*

[1]Department of Molecular and Cell Biology, University of California, Berkeley, Berkeley, United States; [2]Lawrence Berkeley National Laboratory, Berkeley, United States; [3]Department of Ecology and Evolutionary Biology, University of California, Irvine, Irvine, United States; [4]Centre for Chromosome Biology, National University of Ireland, Galway, Ireland

**Abstract** Tandemly-repeated DNAs, or satellites, are enriched in heterochromatic regions of eukaryotic genomes and contribute to nuclear structure and function. Some satellites are transcribed, but we lack direct evidence that specific satellite RNAs are required for normal organismal functions. Here, we show satellite RNAs derived from AAGAG tandem repeats are transcribed in many cells throughout *Drosophila melanogaster* development, enriched in neurons and testes, often localized within heterochromatic regions, and important for viability. Strikingly, we find AAGAG transcripts are necessary for male fertility, and that AAGAG RNA depletion results in defective histone-protamine exchange, sperm maturation and chromatin organization. Since these events happen late in spermatogenesis when the transcripts are not detected, we speculate that AAGAG RNA in primary spermatocytes 'primes' post-meiosis steps for sperm maturation. In addition to demonstrating essential functions for AAGAG RNAs, comparisons between closely related *Drosophila* species suggest that satellites and their transcription evolve quickly to generate new functions.

*For correspondence:
gkarpen@berkeley.edu

Competing interests: The authors declare that no competing interests exist.

## Introduction

Long arrays of tandemly repeated short DNA sequences (known as satellites) are abundant in centromeres (*Sun et al., 2003*) and pericentromeric regions (*Hoskins et al., 2007*), and contribute to chromosome segregation and other heterochromatin functions (*Dernburg et al., 1996*; *Ferree and Barbash, 2009*). Surprisingly, satellite DNAs are expressed in many multicellular eukaryotes, and their aberrant transcription may contribute to carcinogenesis and cellular toxicity (*Yap et al., 2018*; *Jain and Vale, 2017*; *Zhu et al., 2011*). Satellite transcription and/or RNAs may also promote centromere and heterochromatin functions (*McNulty et al., 2017*; *Johnson et al., 2017*; *Velazquez Camacho et al., 2017*; *Shirai et al., 2017*; *Rošić et al., 2014*).

In *D. melanogaster*, simple, tandemly repeated satellite DNAs, such as AAGAG(n) and AATAT(n), comprise ~15–20% of the genome (*Lohe et al., 1993*; *Hoskins et al., 2015*). Given the emerging roles of non-protein coding RNAs (ncRNAs) in chromatin organization and other biological functions (*Rinn and Chang, 2012*), we investigated whether heterochromatic satellite transcripts are required for normal viability and development.

## Results

We first analyzed RNA expression for 31 of the most abundant satellite DNAs, using published RNA-seq data (modENCODE) (*Brown et al., 2014*) and RNA-Fluorescence In-Situ Hybridization (RNA-FISH) (*Figure 1—figure supplement 1*). Further characterizations and functional analyses were focused on AAGAG(n) RNA (hereafter AAGAG RNA) because it is highly abundant, and a previous study suggested it was linked to the nuclear matrix and necessary for viability (*Pathak et al., 2013*). Northern blot analysis of RNA isolated from stage 1–4 embryos shows that AAGAG RNA is maternally loaded as an ~1500 nucleotide (nt) transcript. Smaller RNAs (~20–750 nt) accumulate in later stage embryos (2–24 hr) and third instar larvae (L3 larvae) (*Figure 1—figure supplement 2A*). AAGAG RNA-FISH in 0–18 hr embryos and L3 larvae revealed localization to only one or a few nuclear foci, with no visible cytoplasmic signal (*Figure 1*, A and D). AAGAG RNA foci are not detected prior to embryonic cycle 11, but by cycles 12 and 13, 33% and 67% of embryos (respectively) have one or more foci (*Figure 1—figure supplement 2, B and C*). Furthermore, 100% of embryos exhibit nuclear AAGAG RNA foci by blastoderm (cycle 14,~2 hr after egg laying), coincident with the formation of stable, mature heterochromatin (*Strom et al., 2017*; *Yuan and O'Farrell, 2016*) (*Figure 1A* and *Figure 1—figure supplement 2D*). Surprisingly, the complementary RNA (CUCUU(n)) is not observed in Northern or RNA-FISH analysis (*Figure 1—figure supplement 4, B* and data not shown, respectively), suggesting that most or all of the stable embryo RNA expressed from tandem AAGAG(n) DNA present at multiple genome locations corresponds to AAGAG(n) and not CUCUU(n). This conclusion is supported by the results of RNase digestion experiments, which demonstrate that cycle 14 AAGAG RNA foci contain single-stranded RNA (ssRNA), and not R-loops or double-stranded RNA (dsRNA) (*Figure 1—figure supplement 3*). A combination of transcriptome mining, Northern blotting and RNA-FISH indicates that the majority of AAGAG RNA is transcribed from loci in 2R, X and 3R heterochromatin (*Figure 1—figure supplement 4*). Finally, we ruled out the possibility that detected foci represent DNA, since signal was abolished by RNaseIII, but not RNaseH treatments after probe hybridization (*Figure 1—figure supplement 5*).

To determine where these transcripts localize within the nucleus, we simultaneously performed antibody staining (IF) for a histone post-translational modification enriched in heterochromatin (H3K9me3), and FISH for both AAGAG RNA and DNA. In cycle 12 embryos, AAGAG RNA is distributed randomly throughout the nucleus (*Figure 1E*) and does not co-localize with AAGAG(n) DNA. Once stable heterochromatin forms (cycle 14) (*Yuan and O'Farrell, 2016*), AAGAG RNA foci specifically co-localize with H3K9me3 (*Figure 1E*). By stage 13 embryos (~9.5 hr after egg-laying) AAGAG RNA is specifically enriched in the ventral ganglia (neural tissue), and foci remain either co-localized with or immediately adjacent to heterochromatin (*Figure 1B and D*). In addition, AAGAG RNA localizes to the chromocenter in polytene larval salivary glands (*Figure 1C*).

The presence of AAGAG RNA throughout development suggested a potential role in development or viability. This hypothesis was tested by depleting AAGAG RNA in somatic cells, using actin-GAL4-driven AAGAG shRNA expression (*Figure 1—figure supplement 6*). Depletion of AAGAG RNA results in significantly lower viability by pupal stage compared to controls, with most lethality occurring during third instar larval (L3) stages (*Figure 1—figure supplement 6, G and H*, respectively). We conclude that AAGAG RNA associates with the earliest forms of heterochromatin, maintains this localization at least partially throughout embryonic and larval development, is enriched in neural tissue, and is important for viability.

Surviving act-GAL4-driven AAGAG RNAi adults exhibited partial sterility, prompting further investigation into the distribution and potential functions of AAGAG RNA in the germ line (see *Figure 2—figure supplement 1* for an overview of spermatogenesis). In larval and adult testes, high levels of AAGAG RNA are observed in primary spermatocytes, where they are enriched in regions adjacent to the DAPI-bright 'chromosome territories' located at the nuclear periphery (*Figure 2*, A to C). This pattern is distinct from CUCUU(n) RNA, which is localized to the lumen in primary spermatocytes (*Figure 2—figure supplement 3*). AAGAG RNA is not detectable, even with amplified signal, at earlier stages near the hub, or at later stages (meiosis I and II, and subsequent stages of sperm development). Spermatocyte AAGAG RNA originates from the same 2R, 3R and X heterochromatic satellite regions identified in somatic cells and is specifically not generated from the Y chromosome (*Figure 2—figure supplement 2A and B*, respectively).

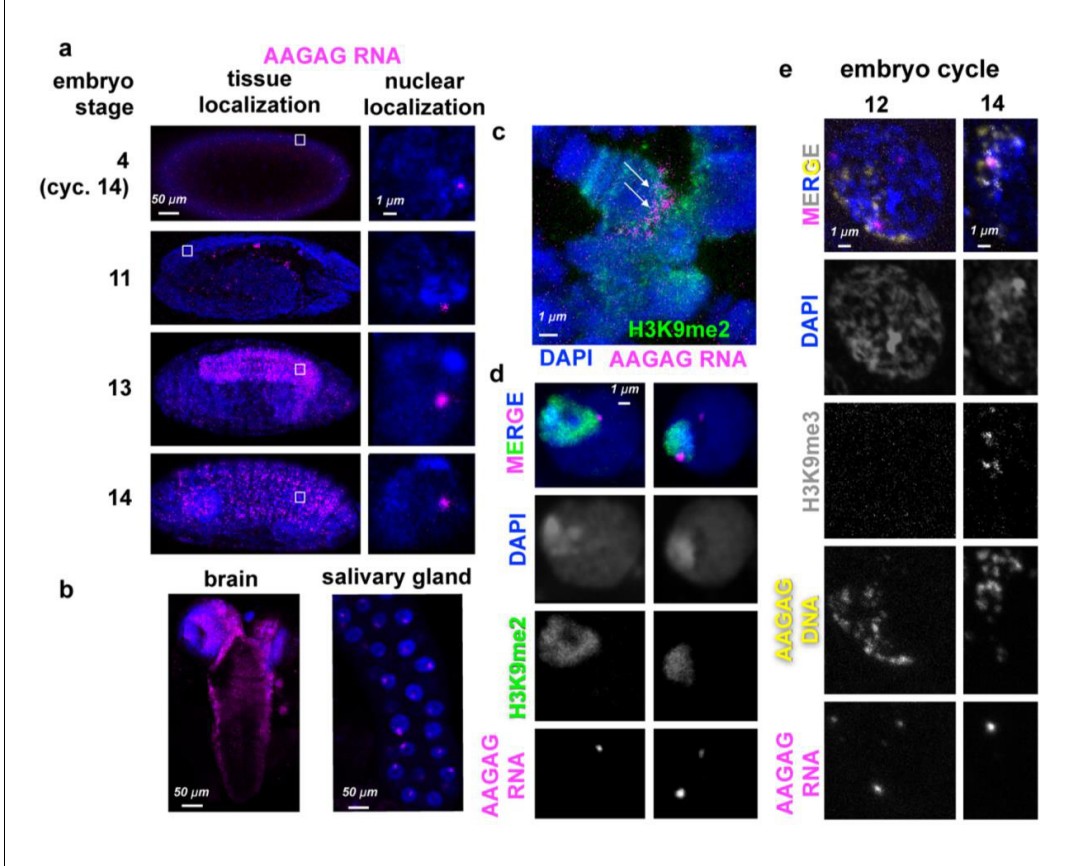

**Figure 1.** AAGAG(n) RNA localizations in embryos and larvae. (**a**) AAGAG RNA distributions (magenta) throughout embryonic and larval development in Oregon R flies. DNA/DAPI = blue; all images are confocal sections. White box indicates location of enlarged nucleus (right column). (**b**) Distributions of AAGAG RNA in intact larval L3 brain (left) and salivary gland (SG) tissue (right) (confocal sections). (**c**) Salivary gland squash projection indicating presence of AAGAG RNA (magenta, see arrows) at the chromocenter (marked with H3K9me2), and not the euchromatic arms. (**d**) Brain cell sections show that there are one or two AAGAG RNA foci per nucleus that are located in or near the pericentromeric heterochromatin (H3K9me2 antibody IF, green). Specifically, 100% of nuclei (N = 5) with AAGAG foci contain foci that completely or partially co-localize with H3K9me2 (left panel). Of these nuclei, (20%) have an additional AAGAG focus that generally does not co-localize with H3K9me2. (**e**) Projections of representative nuclei probed for AAGAG RNA (magenta) and AAGAG DNA (yellow) and stained for H3K9me3 (gray) and DNA (DAPI = blue). Left = cycle 12 nuclei prior to stable heterochromatin formation; right = early cycle 14 nucleus during heterochromatin formation. Note that in cycle 12, the few AAGAG RNA foci do not co-localize with AAGAG DNA. In cycle 14, AAGAG RNA foci co-localize with AAGAG DNA and H3K9me3.

The online version of this article includes the following figure supplement(s) for figure 1:

**Figure supplement 1.** RNA-FISH analysis of satellite RNAs in cycle 14 embryos.

**Figure supplement 2.** AAGAG RNA is present throughout development and forms foci.

**Figure supplement 3.** AAGAG RNA foci contain single-stranded RNA and are not associated with R-loops.

**Figure supplement 4.** AAGAG RNA transcripts originate from 2R, X and 3R heterochromatin loci and are transcribed in embryos and larval brain.

**Figure supplement 5.** AAGAG RNA-FISH localizes RNA and not DNA.

**Figure supplement 6.** AAGAG RNA is decreased and foci abolished in L3 with actin-GAL4-driven RNAi to AAGAG, without affecting levels of genes whose mRNAs contain short runs of AAGAG.

To deplete AAGAG RNA in 4–16 cell spermatogonial cysts, we used the Bag of marbles (Bam)-GAL4 (*White-Cooper, 2012*) driver to express AAGAG shRNA. Strikingly, AAGAG depletion (~72% reduction) results in 100% male sterility, with no impact on female fertility (*Figure 2D*). AAGAG RNAi using drivers expressed earlier in spermatogenesis does not cause fertility defects (*Table 1*). We conclude that expression of AAGAG RNA in primary spermatocytes is required for male fertility.

These results suggested that male infertility upon AAGAG RNA depletion would be caused by defects at stages where AAGAG RNA is expressed. Surprisingly, Bam-GAL4-driven depletion of AAGAG RNA resulted in no gross morphological defects prior to or during meiosis I or II in pupal or

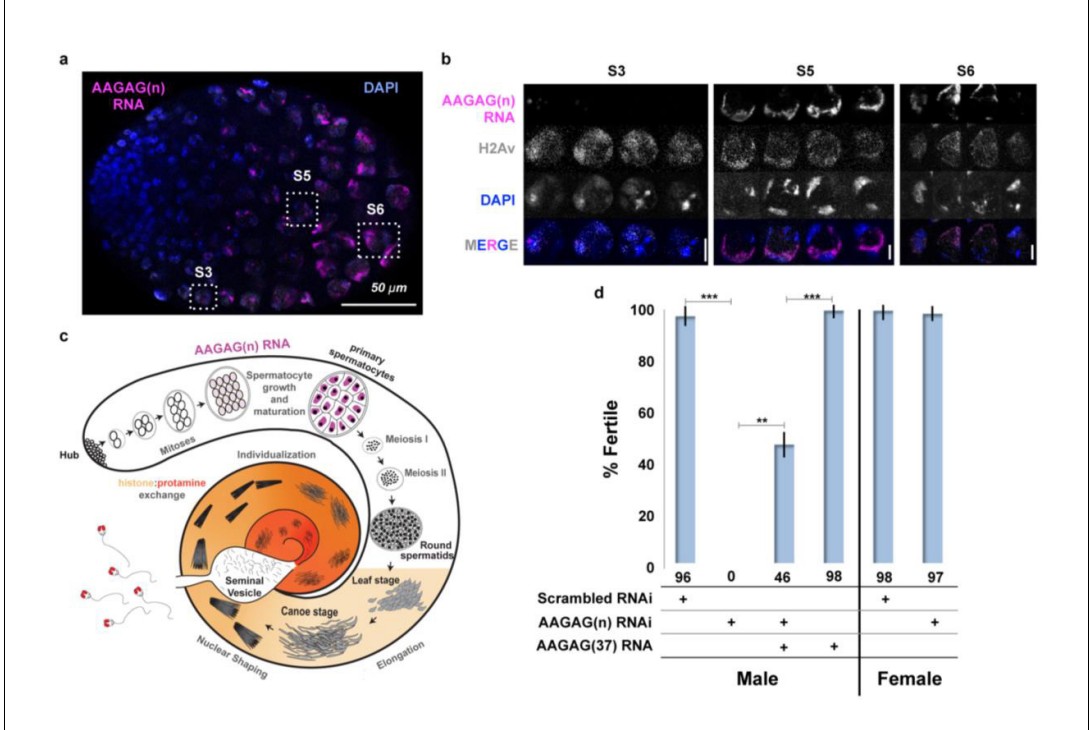

**Figure 2.** AAGAG RNA is enriched in primary spermatocytes and necessary for male fertility. (a) Confocal section of a larval testis. RNA-FISH to AAGAG = magenta, H2Av (chromatin) IF = gray, DNA (DAPI) = blue. S3, S5, and S6 refer to primary spermatocyte stages. (b) Enlarged confocal sections (representative boxes in a) of spermatocyte stages in larvae testes; scale bars = 5 μm. (c) Schematic summary of AAGAG RNA (magenta) localization in adult testes (see *Figure 2—figure supplement 1* for a detailed description of spermatogenesis stages and events). AAGAG RNAs are visible in 16 cell primary spermatocytes (dark pink), and potentially 16 cell spermatogonial cysts (light pink); no AAGAG RNA was detected at earlier stages (hub, 2–8 cell spermatogonial cysts) or after the primary spermatocyte stage (meiosis I and II, sperm elongation- which includes leaf, canoe, individualization steps, and maturation). Post-round spermatid stages are indicated as spermatid nuclei. (d) Fertility after depletion of AAGAG(n) RNA in male primary spermatocytes or female ovaries using the Bam-GAL4 driver. An ~72% reduction in AAGAG RNA levels in testes (see *Figure 2—figure supplement 3, B and C*) results in complete male sterility but has no effect on female fertility. Expression of AAGAG(37) RNA simultaneously with AAGAG RNAi (both driven by Bam-Gal4) partially rescues male sterility (46% fertile). Expression of AAGAG RNA alone, without depletion of endogenous AAGAG RNAs, has no impact on male fertility. Statistically significant differences based on T-tests (two tailed, type three) are indicated by horizontal lines; ***p<0.001, **p<0.01; variation is represented by stdev.

The online version of this article includes the following figure supplement(s) for figure 2:

**Figure supplement 1.** Overview of normal spermatogenesis and defects observed after AAGAG RNA depletion.

**Figure supplement 2.** Heterochromatic regions adjacent to AAGAG(n) or AG(n)-rich blocks are transcribed in primary spermatocytes, co-localize with AAGAG(n) RNA foci and do not come from the Y.

**Figure supplement 3.** AAGAG RNA and not CUCUU RNA is substantially decreased in Bam-GAL4- driven AAGAG RNAi, and AAGAG RNA levels are increased in rescue experiments.

**Table 1.** Male fertility in AAGAG RNAi with GAL4 drivers expressed at earlier testes stages than Bam.

| GAL4 RNAi driver | Expression location (*Demarco et al., 2014*) | % fertile | + /- stdev. | Minimum number of males per set |
|---|---|---|---|---|
| Fascillin | Hub | 94 | 16 | 15 |
| PTC | Soma- CySCs and cyst cells | 90 | 5 | 18 |
| Traffic Jam | Soma- Hub and CySCs | 97 | 4 | 12 |
| Dpp1 | Soma- CySCs and early cyst cells | 96 | 6 | 17 |
| Nanos | Germline- GSCs and early germline cysts | 83 | 5 | 13 |

adult (0–6 hr and 4–7 days post-eclosion) testes. However, individualized mature sperm DNA was completely absent from the seminal vesicles (SV), in contrast to their abundance in controls (*Figure 3A*), demonstrating that AAGAG RNA is important for later steps in spermatogenesis. In fact, the first visible defects are observed during the canoe, individualization and maturation stages (*Figure 3—figure supplement 1A* and *Figure 3B*), which are devoid of detectable AAGAG RNA in wild-type testes (*Figure 2C*). For instance, aberrant canoe stage and individualizing sperm DNA (i.e. irregular, long and decondensed sperm DNA) were observed at significantly higher frequencies after AAGAG RNA depletion, compared to scrambled RNAi controls (*Figure 3—figure supplement 1* and *Figure 3E*). At later individualization stages, sperm bundles in AAGAG RNA depleted testes often contained less than the normal 64 sperm and were disorganized, displaying 'lagging' sperm nuclei and loosely packed sperm bundles (*Figure 3, B and E*). Finally, sperm DNA present was abnormally 'kinked,' 'needle eyed' or 'knotted' in appearance, and normal, mature forms of sperm

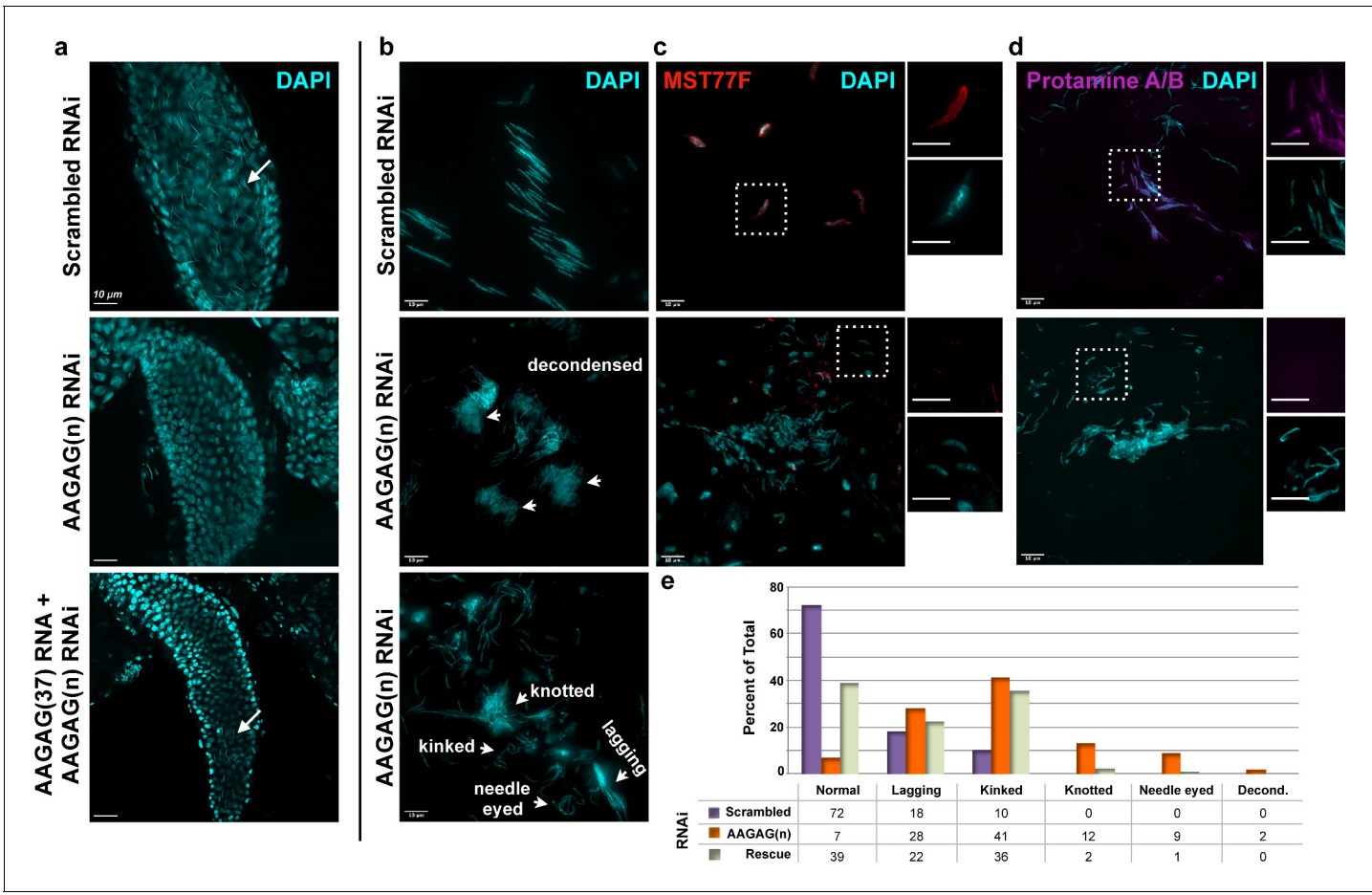

**Figure 3.** AAGAG RNAi depletion in mitotic germline cysts and spermatocytes (Bam-GAL4 driver) results in severe defects in sperm maturation and protamine deposition. (a) Seminal vesicles (SVs) in testes from 0 to 6 hr old adults; DAPI (DNA) = cyan. Mature sperm nuclei visible as thin, elongated DAPI signals in the scrambled control (top, white arrow) are absent after AAGAG RNAi. Individualized mature sperm (white arrow) are visible in SVs from AAGAG RNAi males that also express AAGAG(37) RNA (partial rescue, 4–7 day old adults). (b) Bundles of elongating sperm nuclei visible in the scrambled RNAi control (top). Defective 'decondensed' (middle, white arrowheads), 'knotted,' 'kinked 'needle eyed' and 'lagging' (bottom, white arrowheads) sperm phenotypes are visible in the AAGAG RNAi but are much less frequent or absent in controls (see e). (c) Transition Protein Mst77F (red) is present on sperm DNA in control RNAi but is largely absent and/or disorganized after AAGAG RNAi (dashed boxes indicate regions in the zoomed images to the right). (d) Protamine A/B (purple) is present on sperm DNA in the scrambled control RNAi but is absent after AAGAG RNAi. Scale bars = 10 μm except for zoomed images in c and d = 8 μm. (e) Quantitation of sperm defects (4–6 day adult testes) associated with AAGAG RNAi depletion, along with AAGAG RNA rescue, compared to scrambled RNAi control.

The online version of this article includes the following figure supplement(s) for figure 3:

**Figure supplement 1.** Histones are retained and DNA morphology is altered in late canoe stage AAGAG RNAi testes.

DNA readily found in basal regions (just prior to entry into the seminal vesicle) of control testes were never observed after AAGAG depletion (*Figure 3B*). These phenotypes indicated that AAGAG RNA is important for sperm nuclear organization, similar to the consequences of defective histone-protamine transitions observed previously (*Rathke et al., 2010*; *Jayaramaiah Raja and Renkawitz-Pohl, 2006*). Strikingly, antibody IF revealed that Bam-GAL4-driven AAGAG RNA depletion caused reduced and defective incorporation of the transition protein Mst77F (*Figure 3C*), an absence of Protamine A/B (*Figure 3D*), and histone retention into the late canoe stage (*Figure 3—figure supplement 1*).

Importantly, fertility defects resulting from AAGAG RNA depletion are partially rescued by simultaneously expressing AAGAG RNA (185 bases, 37 repeats), when both are controlled by the Bam-GAL4 driver. Under these conditions we observe a 2-fold increase in AAGAG RNA signal compared to AAGAG RNAi alone (*Figure 2—figure supplement 3D*), which is sufficient to partially restore male fertility (46% with AAGAG RNA expression compared to 0% in AAGAG RNAi alone, *Figure 2D*), the presence of mature sperm in the seminal vesicles (*Figure 3A*), and normal sperm DNA morphology (*Figure 3E* and *Figure 3—figure supplement 1B*). We conclude that RNA transcribed from the simple tandem repeat AAGAG(n) in primary spermatocytes is necessary for completing spermatogenesis and male fertility in *Drosophila melanogaster*, at least in part by promoting the histone-protamine transition and/or other post-meiotic steps in sperm maturation.

## Discussion

Here, we demonstrate that AAGAG(n) satellite RNAs are transcribed from heterochromatic regions on multiple chromosomes, cluster into nuclear foci, associate with the earliest forms of heterochromatin in embryos, and persist throughout fly development. AAGAG RNA is important for viability, though further investigations are necessary to determine its functions in early development. Most strikingly, we observe that AAGAG RNA is expressed in the male germ-line and is absolutely essential for male fertility.

It is surprising that AAGAG RNA is expressed only in primary spermatocytes yet is critical for completing much later stages of sperm development, when AAGAG RNA is not detected. Specifically, defects in late spermatogenesis, including canoe, individualization and maturation stages and the histone-protamine exchange, were observed when AAGAG RNA was depleted in primary spermatocytes, and expression of AAGAG RNA at the same stage partially restored these fertility and sperm defects. It is interesting that aberrant histone-protamine transition and sperm individualization are also observed in Segregation Distorter (SD) testes, where the affected sperm contain abnormally high numbers of another satellite repeat (Responder, or Rsp) (*Larracuente and Presgraves, 2012*). We suggest that AAGAG RNA, and perhaps other satellite RNAs (e.g. Rsp), function in primary spermatocytes to 'prime' cells and/or chromosomes to successfully accomplish downstream, post-meiotic sperm development.

Although the molecular mechanisms directly impacted by AAGAG RNA are currently unknown, the spatial and temporal disconnect between its expression and depletion phenotypes limit the possibilities. We speculate that proper histone:protamine exchange and post-meiotic chromatin organization require AAGAG RNA in primary spermatocytes to sequester or exclude factors that regulate localization of late-acting proteins or ncRNAs (*Figure 4A*), form essential complexes or alter post-translational modifications (*Figure 4B*), or regulate global genome organization (*Figure 4C*), such as condensation or chromosome 'bundling' (*Jagannathan et al., 2019*), which could impact expression of genes critical for later spermatogenesis events. It is also possible that AAGAG RNA directs the proper chromatin organization of the cognate satellite DNAs (*Figure 4D*), as demonstrated for small RNA-directed, homology-based recruitment of histone modifying proteins to heterochromatin (*Allshire and Madhani, 2018*).

It is also worth noting that the expression of simple repeats for essential functions seems incompatible with the fast evolution of satellite DNAs, reflected in dramatic changes in both sequences and copy numbers across species (*Wei et al., 2018*). Specifically, AAGAG is one of the most abundant simple repeats in *D. melanogaster*, comprising ~5% of the genome (*Lohe and Brutlag, 1986*). However, the amount of AAGAG is several orders of magnitude lower in the closely related *D. simulans* and *D. sechellia*, and is nearly absent in other *Drosophila* species (*Wei et al., 2018*). It is possible that in species with few or no AAGAG repeats, low levels of AAGAG RNA are sufficient for

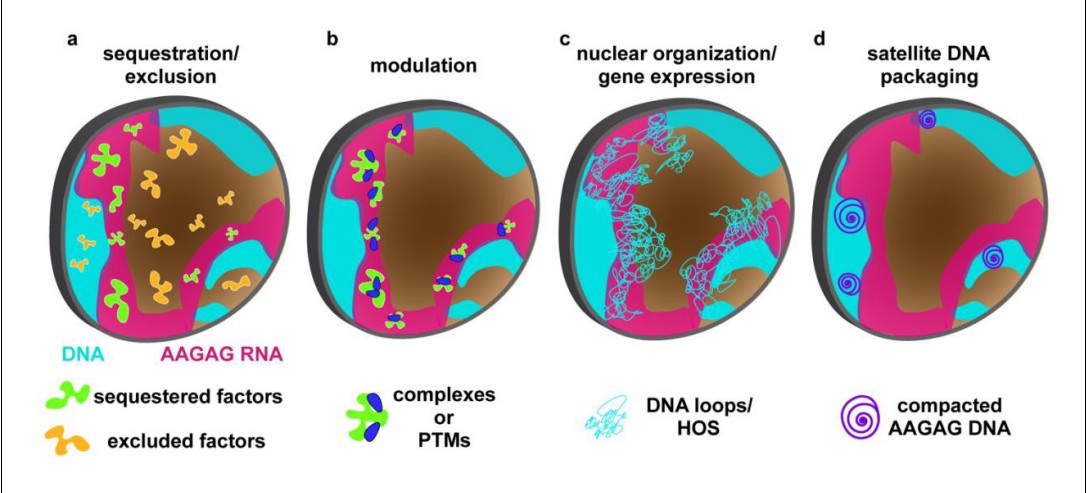

**Figure 4.** Model for AAGAG RNA function during spermatogenesis. AAGAG RNA (magenta) present only in primary spermatocytes (light blue = chromosome territories) acts directly or indirectly to promote important processes later in sperm maturation, including the histone-protamine transition and individualization. AAGAG RNA could ensure normal completion of later events by mediating: (a) proper localization of factors (RNA and/ or protein) through sequestration (green) or exclusion (orange), (b) formation of molecular complexes or modifications (e.g. PTMs) (green blobs plus blue ovals), (c) regulation of global DNA/chromatin organization (e.g. condensation, Y loops, Higher Order Structures (HOS)) which for example could impact expression of critical spermatogenesis genes, or (d) local DNA/chromatin organization of cognate AAGAG loci, as observed for heterochromatin recruitment by siRNAs. Although direct experiments are required to test these models, we favor d) because it can accommodate both fast turnover of satellite sequences during evolution and sequence-independent roles in ensuring fertility (see text).

fertility, but we favor the hypothesis that expression of different lineage-specific satellite arrays are required for normal sperm maturation. In this context, it is interesting that new lineage-specific protein-coding genes (*Chen et al., 2013*) are biased toward testis-expression and acquisition of essential functions in male reproduction, including spermatogenesis (*Ding et al., 2010*). Selective pressures proposed to drive the fast evolution of new testis-expressed genes could also impact satellite RNA evolution and function, such as sperm competition, sexual conflict, or antagonistic interactions with germline parasites and/or selfish DNAs (*Kaessmann, 2010*) . However, it is unclear how completely different satellite RNA sequences would retain functions such as promoting formation or proper localization of regulatory complexes required for later spermatogenesis events (*Figure 4A– C*). Thus, we posit that a requirement for satellite RNA-mediated packaging of cognate satellite DNAs (*Figure 4D*) provides the most parsimonious explanation for both the fast turnover and its roles in ensuring fertility. This model is attractive because transcription of any new or even evolving satellites would avoid deleterious dis-organization of the corresponding DNAs, independent of RNA primary sequences or secondary structures. Detailed analyses of the functions of distinct satellite RNAs in *D. melanogaster* and other Drosophilds are required to test the mechanistic hypotheses outlined in *Figure 4*. Regardless, our results provide a strong impetus for additional studies of satellite RNA functions, which could elucidate new roles of so-called 'junk DNA' in health, disease and evolution.

## Materials and methods

### Imaging

Most images were acquired using a Zeiss LSM710 confocal microscope using 40X water or 63X oil objectives. For these confocal images, projections were acquired as z-stacks with step sizes depending on the sample. Image files were then processed and analyzed using Fiji. Non-rescue testes images in *Figure 3* and *Figure 3—figure supplement 1* were acquired using DeltaVision Elite wide-field microscope system (Applied Precision). Images were acquired as z-stacks with a step size of 0.5 μm, raw data files were deconvolved using a maximum intensity algorithm. 3D z-stack images were represented in 2D by projection using SoftWorx (Applied Precision).

## RNA probe generation for RNA-FISH

RNA probes were made by using oligo templates with antisense T3 promoters on the 3'ends, hybridizing an oligo composed of sense T3 promoter so as to create a double stranded 3' end, or in the case of 359 bp repeat, amplification with oligos containing T3 and T7 promoter ends on genomic DNA using standard protocols. Probe templates were then transcribed with T3 RNA polymerase (or T7 for one strand of 359 bp repeat) and either UTP-biotin or UTP-digoxigenin labels, or in the case of RNA without Uracil, biotin-ATP. Oligos are listed in *Table 2* and were ordered standard desalted from IDT. Reaction conditions were as follows: In a 40 ul reaction, 1X RNApol reaction buffer (NEB cat. MO3782), 1 mM each final concentration of ATP, GTP, CTP and 0.62 mM UTP, supplemented with 0.35 mM final concentration of either digoxegenin-11-UTP (Roche cat. 3359247910), biotin-UTP (Sigma, cat. 11388908910), or biotin-11-ATP (Perkin Elmer, cat. NEL544001EA), 1 Unit Protector RNase inhibitor (Roche cat. 3335402001), 5 µM each of probe template and T3 promoter oligo (5'-AATTAACCCTCACTAAAG), and $H_20$ to 40 µl were combined. Reactions were heated to 80°C, 3 min to denature probes, iced 2 min., 4 µl (or 200Units) of T3 (or T7) RNA polymerase (NEB cat. M0378S) added and incubated at 37°C overnight. 2 µl Turbo DNAse (ThermoFisher Cat. AM2238) was then added to degrade DNA templates, incubated at 37°C for 15 min and the reaction stopped by adding 1.6 µl of 500 mM EDTA. Probes were then purified using standard sodium acetate/ethanol purification. Probe concentration was then assessed using Qubit RNA high sensitivity protocols and reagents and stored at −80°C.

## RNA-FISH buffers

PBT solution: 1X PBS and 0.1% Tween-20.
Western Blocking Reagent 10X: 10% casein in 100 mM maleic acid; 150 mM NaCl; pH 7.5. heated at 60°C for 1 hr to dissolve.
PBT block: 1:1 PBT/2X WBR;
Hybridization buffer: 50% formamide, 5X SSC, 100 µg/mL heparin, 100 µg/mL sonicated salmon sperm DNA, and 0.1% Tween-20, filtered through a 0.2 µm filter.

For clarity, the methods for RNA-FISH probe hybridization and detection are numbered below.

## RNA-FISH methods

### Protocol 1. RNA-FISH probe hybridization and primary antibody incubation

RNA probe hybridization for all tissues was carried about according to *Legendre (2013)*, steps 10–17 under subheading #3. Samples were then washed one time with PBT then blocked in PBT block 1 hr at room temperature. Samples were then processed for either 'non-Tyramide Signal Amplification (TSA) probe amplification' (Protocol 2) or 'TSA amplification for RNA-FISH probe detection' (Protocol 3).

### Protocol 2. Non-TSA probe detection for RNA-FISH

For 'non-TSA amplification', samples were incubated with either mouse anti-digoxigenin-Cy5 (source unknown) or rabbit anti-digoxigenin A488 (Invitrogen cat# 700772) in PBT block at 1/200 dilution for 1 hr at room temperature. Afterwards, samples were incubated 6x's 10 min each in PBT block, stained with DAPI 10 min, washed 3'xs 10 min. each in PBS and mounted in Prolong-Gold antifade mountant (Thermofisher, cat. P36390).

#### Figures processed using this protocol

*Figure 1D* (see below for 1C); *Figure 2A and B*; *Figure 1—figure supplement 4, C–H* for AAGAG only; *Figure 1—figure supplement 6, A–C*; *Figure 2—figure supplements 2* and *3*.

### Protocol 3. TSA amplification for RNA-FISH probe detection

For samples undergoing 'TSA amplification for RNA-FISH probe detection,' samples were incubated with primary antibody (1/400 dilution of mouse anti-digoxigenin coupled to biotin (Jackson Immuno Research cat.200-062-156, lot. 123482)), with 0.2 U/µl protector RNAse inhibitor and incubated overnight at 4°C. Next, samples were washed 6x's 10 min each in PBT block. The next steps are essentially as per 'tyramide signal amplification kit' protocols (ThermoFisher) but with reagents purchased separately: Samples were incubated with 1:100 streptavidin-HRP (Molecular probes, cat. S911) in

**Table 2.** Oligos for RNA probes.

| Repeat or region | Oligo with T3 antisense promoter |
|---|---|
| CAGC(n) | CAGCCAGCCAGCCAGCCAGCCAGCTCTCCCTTTAGTGAGGGTTAATT |
| CCCA(n) | CCCACCCACCCACCCACCCACCCACCCATCTCCCTTTAGTGAGGGTTAATT |
| CATTA(n) | CATTACATTACATTACATTATCTCCCTTTAGTGAGGGTTAATT |
| CGGAG(n) | CGGAGCGGAGCGGAGCGGAGCGGAGTCTCCCTTTAGTGAGGGTTAATT |
| CGA(n) | CGACGACGACGACGACGACGACGATCTCCCTTTAGTGAGGGTTAATT |
| CAACT(n) | CAACTCAACTCAACTCAACTCAACTTCTCCCTTTAGTGAGGGTTAATT |
| CGAAG(n) | CGAAGCGAAGCGAAGCGAAGCGAAGTCTCCCTTTAGTGAGGGTTAATT |
| CCCCAG(n) | CCCCAGCCCCAGCCCCAGCCCCAGTCTCCCTTTAGTGAGGGTTAATT |
| CCGAG(n) | CCGAGCCGAGCCGAGCCGAGCCGAGTCTCCCTTTAGTGAGGGTTAATT |
| CGGAA(n) | CGGAACGGAACGGAACGGAACGGAATCTCCCTTTAGTGAGGGTTAATT |
| CACCC(n) | CACCCCACCCCACCCCACCCCACCCTCTCCCTTTAGTGAGGGTTAATT |
| CTAGT(n) | CTAGTCTAGTCTAGTCTAGTCTAGTTCTCCCTTTAGTGAGGGTTAATT |
| CATCG(n) | CATCGCATCGCATCGCATCGCATCGTCTCCCTTTAGTGAGGGTTAATT |
| CAT(n) | CATCATCATCATCATCATCATCATTCTCCCTTTAGTGAGGGTTAATT |
| CAAAC(n) | CAAACCAAACCAAACCAAACCAAACTCTCCCTTTAGTGAGGGTTAATT |
| CGAAA(n) | CGAAACGAAACGAAACGAAACGAAATCTCCCTTTAGTGAGGGTTAATT |
| CATAT(n) | CATATCATATCATATCATATCATATTCTCCCTTTAGTGAGGGTTAATT |
| GAAA(n) | GAAAGAAAGAAAGAAAGAAAGAAATCTCCCTTTAGTGAGGGTTAATT |
| CAGAA(n) | CAGAACAGAACAGAACAGAACAGAATCTCCCTTTAGTGAGGGTTAATT |
| AAGGAG(n) | AAGGAGAAGGAGAAGGAGAAGGAGAAGGAGTCTCCCTTTAGTGAGGGTTAATT |
| AAGAGG(n) | AAGAGGAAGAGGAAGAGGAAGAGGAAGAGGTCTCCCTTTAGTGAGGGTTAATT |
| AATAC(n) | AATACAATACAATACAATACAATACAATACTCTCCCTTTAGTGAGGGTTAATT |
| AATAG(n) | AATAGAATAGAATAGAATAGAATAGAATAGTCTCCCTTTAGTGAGGGTTAATT |
| AATAGAC(n) | AATAGACAATAGACAATAGACAATAGACTCTCCCTTTAGTGAGGGTTAATT |
| AATAACATAG(n) | AATAACATAGAATAACATAGAATAACATAGTCTCCCTTTAGTGAGGGTTAATT |
| AACAC(n) | AACACAACACAACACAACACAACACAACACTCTCCCTTTAGTGAGGGTTAATT |
| dodeca(n) | ACCGAGTACGGGACCGAGTACGGGTCTCCCTTTAGTGAGGGTTAATT |
| GTGTT(n) | GTGTTGTGTTGTGTTGTGTTGTGTTGTGTTTCTCCCTTTAGTGAGGGTTAATT |
| GTAAT(n) | GTAATGTAATGTAATGTAATGTAATGTAATTCTCCCTTTAGTGAGGGTTAATT |
| GTATT(n) | GTATTGTATTGTATTGTATTGTATTGTATTTCTCCCTTTAGTGAGGGTTAATT |
| TTAA (n) | TTAATTAATTAATTAATTAATTAATTAATTAATCTCCCTTTAGTGAGGGTTAATT |
| CAAT (n) | CAATCAATCAATCAATCAATCAATCAATCAATTCTCCCTTTAGTGAGGGTTAATT |
| AAGAG(n) | GAGAAGAGAAGAGAAGAGAAGAGAAGAGAAGAGAAGAGAATCTCCCTTTAGTGAGGGTTAATT |
| CTCTT(n) | CTCTTCTCTTCTCTTCTCTTCTCTTCTCTTTCTCCCTTTAGTGAGGGTTAATT |
| 359 Forward | AATTAACCCTCACTAAAGGGAGAAATGGAAATTAAATTTTTTGG |
| 359 Reverse | TTAATACGACTCACTATAGGGAGAGTTTTGAGCAGCTAATTACC |
| chr2R:1,825,641–1825699 sense | GGCAGTTTATGTGCGTACAACAACAACAGGACTGCAAACAAAACACGAAACAGATATTTTTCTCCCTTTAGTGAGGGTTAATT |
| chr2R:1,825,641–1825699 anti-sense | AAAATATCTGTTTCGTGTTTTGTTTGCAGTCCTGTTGTTGTTGTACGCACATAAACTGCCTCTCCCTTTAGTGAGGGTTAATT |
| chr2R:1,826,691–1,826,740 sense | TAGACACATCTACGAAGACACAATTCTACAAGAACTAAACAACAAAAAGTTCTCCCTTTAGTGAGGGTTAATT |
| chr2R:1,826,691–1,826,740 anti-sense | ACTTTTTGTTGTTTAGTTCTTGTAGAATTGTGTCTTCGTAGATGTGTCTATCTCCCTTTAGTGAGGGTTAATT |
| chrX:11,830,844–11,830,910 sense | CCAAGCTTCAGGAGAAAGAGAAAGAAGAAAGCTTTAAACTTAAGGAAAGAGAAGAGAGCCTTAGGATTCTCCCTTTAGTGAGGGTTAATT |

*Table 2 continued on next page*

*Table 2 continued*

| Repeat or region | Oligo with T3 antisense promoter |
| --- | --- |
| chrX:11,830,844–11,830,910 antisense | CTAAGGCTCTCTTCTCTTTCCTTAAGTTTAAAGCTTTCTTCTTTCTCTTTCTCCTGA AGCTTGGCTTTCTCCCTTTAGTGAGGGTTAATT |
| chrX:12,660,096–12,660,145 sense | TCGCACACACACACGCAACACTTAGGCACACATAGGAGATAGAGTGAGATCTCCCTT TAGTGAGGGTTAATT |
| chrX:12,660,096–12,660,145 anti-sense | TCTCACTCTATCTCCTATGTGTGCCTAAGTGTTGCGTGTGTGTGTGCGATCTCCCTTT AGTGAGGGTTAA TT |
| chrX:22,453,019–22,453,076 sense | CGACAGACAGTAAAATTAAACAAACTGCGGACGCGTGTGACAGAACTAATCCAACTTT CTCCCTTTAGTGAGGGTTAATT |
| chrX:22,453,019–22,453,076 anti-sense | AAGTTGGATTAGTTCTGTCACACGCGTCCGCAGTTTGTTTAATTTTACTGTCTGTCGT CTCCCTTTAGTGAGGGTTAATT |
| chr3R:3,169,758–3,169,820 antisense | TCGGAAGAGACTAAACTTGTGCATTCGATATAGCTCTTTGTCGGCCCTAGCTGCTGTA AACAATCTCCCTTTAGTGAGGGTTAATT |
| chr3R:3,169,758–3,169,820 sense | TTGTTTACAGCAGCTAGGGCCGACAAAGAGCTATATCGAATGCACAAGTTTAGTCTCT TCCGATCTCCCTTTAGTGAGGGTTAATT |
| chr3R:3,170,372–3,170,441 antisense | TTAAACTATATTAAACATTGTATATAAGTATAATAGCGAATACTATTTACGTATATGTTCT TTCATAAATTCTCCCTTTAGTGAGGGTTAATT |
| chr3R:3,170,372–3,170,441 sense | ATTTATGAAAGAACATATACGTAAATAGTATTCGCTATTATACTTATATACAATGTTTAAT ATAGTTTAATCTCCCTTTAGTGAGGGTTAATT |

PBT block for 1 hr at room temperature. Samples were then washed in 1:1 PBT/2XWBR 6x's 10 min each, once with PBT, and 2x's with PBS. Samples were then incubated with Alexa 647 tyramide (TSA Reagent, Alexa Fluor 647 Tyramide cat. T20951) according to company protocols. Essentially, this consisted of adding 1 µl of 30% hydrogen peroxide to 200 µl tyramide signal kit amplification buffer, then diluting this solution 1/100 in tyramide signal amplification buffer for a final hydrogen peroxide concentration of 0.0015%. This solution was then added to the sample and incubated at room temperature for 1 hr in the dark. Samples were then washed 1x with PBS for 10 min, stained with DAPI for 10 min, washed 4x's with PBS 10 min. each, and mounted in Prolong Gold Antifade mountant.

## Figures processed using this protocol
*Figure 1A and B* (see below for 1E); *Figure 1—figure supplements 1–3*; *Figure 1—figure supplement 4, C–H* for non AAGAG RNA detection (ie 2R and X heterochromatic transcripts); *Figure 1— figure supplement 5*.

## RNA-FISH of repeats in embryos
For RNA-FISH of repeat RNAs, 0–8 hr Oregon R embryos were collected on apple juice plates, dechorionated and processed according to *Legendre (2013)*, as per protocols 1 and 3 above, with the exception of using 37% formaldehyde stock from Sigma (cat. F1635-500ML). For *Figure 1—figure supplement 1*, for non-AAGAG repeat RNAs, at least 50 cycle-14 embryos were imaged. With the exception of AAGAG(n) RNA, we did not quantify the percent of embryos with RNA foci. For *Figure 1—figure supplement 2*, at least 10 embryos prior to cycle 12, at least three embryos for cycles 12 and 13, and hundreds of embryos for cycle 14 were imaged for AAGAG(n) foci.

## Co-IF DNA/RNA-FISH of AAGAG RNA in embryos
(*Figure 1E*). Co-IF RNA/DNA-FISH was performed essentially as described in *Shpiz et al. (2013)*, in which RNA-FISH was performed first, signal detected via tyramide signal amplification, RNAse treatment to remove RNA and prevent DNA-FISH probes binding to RNA, and then DNA-FISH performed. Essentially, RNA-FISH was performed as above, but after tyramide signal amplification (protocols 1 and 3 above) and washing, samples were fixed in 4% formaldehyde. Samples were then washed 3x in PBS 2 min. each. RNA was then removed under the following conditions: In a 50 µl final volume, 1X Shortcut RNaseIII buffer (NEB cat. M0245S), 1.5 ul RNASEIII (neb cat. MO245S), 100 µg/ ml RNaseA final concentration, 1X MnCl2 (NEB cat. MO245S) and water to 50 µl were added and samples incubated overnight at 4°C. Samples were then rinsed 3x's in PBT 5 min each, rinsed in 1:5,

1:1 and 5:1 mixtures of PBT: RNA hybridization solution for 15 min each. Samples were then replaced with hybridization buffer and incubated 15 min. A DNA oligo probe to AAGAG(7) tagged with Alexa5 was then diluted in hybridization buffer to 2.5 ng/µl, denatured at 70°C for 3 min, then left on ice for 2 min. Hybridization solution was removed from the embryos, probe solution added, and the sample denatured at 80°C for 15 min and hybridized overnight at 37°C with nutation. Samples were then washed 2x's with pre-warmed 37°C hybridization buffer 10 min each. Samples were then washed in 3:1, 1:1, 1:3 hybridization buffer:PBT 15 min. each at 37°C. Samples were then washed 2x's in PBT at room temperature 5 min. each. Samples were then stained with DAPI 10 min., washed once in PBS, and mounted in Pro-Long Gold Antifade mountant.

## RNA-FISH in larvae

This protocol is essentially as described in *Jandura et al. (2017)*. All figures containing larval RNA-FISH (*Figure 1B and D*, *Figure 2A and B* and *Figure 1—figure supplement 4, C–H*) used protocol A) and C) below. Those processed for TSA (needed for protocol three above) additionally used B below. For *Figure 1—figure supplement 6, A–C*, at least three brain lobes were imaged.

A. Third instar larvae were dissected in PBS supplemented with 0.2 U/µl Protector RNase Inhibitor. The posterior end of the larvae was removed, then the remaining L3 inverted inside out. The inverted larvae were then transferred to ice cold PBS with 0.2 U/µl RNAse inhibitor. Larvae were then fixed in PBT with 4% formaldehyde for 15 min, then washed 3x, 5 min each with PBT. Larvae were then incubated with 0.1%(vol/vol) DEPC in PBT for 5 min to deactivate endogenous RNAses. Samples were then rinsed 2x's with PBS.

B. Use of TSA amplification in L3 requires removal of endogenous peroxidases and requires the following protocol after DEPC treatment above and rinsing in PBS: In order to quench endogenous peroxidases, samples were incubated in 350 µl (enough to cover all tissue) of 3% $H_2O_2$ in PBS 15 min at room temperature and the tube kept open to prevent gas buildup. Samples were then rinsed 2x with PBT 10 min. each.

C. To all larval samples: Larvae were then permeabilized by incubation in 500 µl cold 80% acetone in water at −20°C 10 min. Samples were then washed 2x, 5 min. per wash with PBT, then post fixed with 4% formaldehyde in PBT for 5 min. Samples were then washed 5x's with PBT 2 min each. Samples were then rinsed with 1:1 PBT/RNA hybridization solution, then with 100% RNA hybridization solution, and then stored in hybridization solution at −20°C until needed. Samples were then processed according to RNA-FISH protocol (protocol one above, under 'RNA FISH methods') for probe hybridization and either (protocol two above, under 'RNA FISH methods') for non-TSA probe or (protocol three above, under 'RNA FISH methods') for TSA amplification.

## RNA-FISH in salivary gland squash

(*Figure 1C*) Larvae were grown, prepped and salivary glands processed as per *Cai et al. (2010)*, rehydrated in 95%, 70%, then 30% ethanol 1 min each, then washed 5 min in PBT (0.1% Triton X-100 (TX100)). Slides were then fixed again in 3.7% formaldehyde in PBT (0.1% TX100), washed 2x 3 min. each in PBT (0.1% TX100), treated with 0.1% DEPC in PBT (0.1% TX100) and washed one time in PBT (0.1% TX100). Sample was then covered with pre-denatured hybridization solution, covered with a coverslip and incubated at 56°C in a sealed hybridization chamber for 2 hr. The probe solution was then created by adding 100 ng probe in 100 µl hybridization solution, heating at 80°C for 3 min., and cooling on ice for 5 min. This probe solution was then added to the sample, a coverslip added and sealed with rubber cement, and incubated overnight at 56°C in a humid box. At 55°C in a coplin jar, slides were then treated in 50% formamide/PBT (0.1% tx100) 1 hr, 25% formamide/PBT (0.1% Tx100) 10 min, then 3x with PBT (0.1%Tx100) 10 min each. Once at room temp, samples were blocked in 1:1 PBT/2xWBR and processed as per larval RNA-FISH using non -TSA probe detection (protocol two above).

## RNAse treatment of embryos

(*Figure 1—figure supplement 3*) For RNAse of embryos prior to probe hybridization: RNA-FISH to AAGAG was performed on embryos pre-treated with RNaseIII (which cleaves dsRNA; *Nicholson, 2014*), RNaseH (which cleaves the RNA strand in RNA/DNA hybrids), RNase I (which non-specifically cleaves ssRNA and dsRNA), and RNaseA (which cleaves adjacent to

pyrimidines, preferentially in ssRNA, and specifically not between purines such as 5'-AGAAGGGAGAAG [*Herbert et al., 2018*; *Kelemen et al., 2000*]). Reaction conditions were as follows: Samples were treated in 50 µl final volume with either RNAseIII: 1X RNAseIII buffer, 1.5 µl Shortcut RNaseIII (New England Biolabs, cat. M0245s), and 1X MnCL$_2$; RNAseH treatment: 1X RNAseH buffer, 1.5 µl RNAseH (New England Biolabs, cat. M0297S); RNAse one treatment: 1X RNAseH buffer, 1.5 µl RNAse1 (Ambion cat. AM2294); RNAse A treatment: 1x RNAseH buffer, 15 µg RNAseA- at 37°C for 5 hr. Samples were then washed 5x's in PBT 2 min each, treated with 0.1% DEPC to deactivate any remaining RNAse, then washed in PBT. Samples were then rinsed in 1:1 mixture of PBT:RNA hybridization solution for 2 min and resuspended in 100% hybridization solution. Samples were then processed as per 'RNA-FISH probe hybridization and primary antibody incubation' (protocol one above) and protocol 'TSA amplification for RNA-FISH probe detection' (protocol three above). For each condition, at least 10 entire embryos were imaged.

## RNAse of embryos after probe hybridization

(*Figure 1—figure supplement 5*). After probe hybridization and washing with PBS, samples were treated in 50 µl final volume for either RNAseIII treatment: 1X RNAseIII buffer, 1.5 µl Shortcut RNaseIII (New England Biolabs, cat. M0245s), and 1X MnCL$_2$ or RNAseH treatment: (1X RNAseH buffer, 1.5 µl RNAseH (New England Biolabs, cat. M0297S) at 37°C for 2 hr. Samples were then blocked with 2x PBT:WBR 1 hr then processed as per protocol 'TSA amplification for RNA-FISH probe detection' (protocol three above). Three embryos treated with RNaseH were imaged, while six treated with RNaseIII were imaged.

## RNA-FISH in adult testes

For analysis of AAGAG RNA in RNAi adult testes (*Figure 2—figure supplement 3*), flies were mated at 29°C and F$_1$ progeny grown at 29°C to mimic conditions used to assess sperm morphological defects. AAGAG RNA was also visualized in RNAi testes grown at 25°C to rule out that temperature affected levels and distribution of AAGAG RNA (not shown). For analysis of AAGAG RNA in Oregon R and XO/XY testes (*Figure 2—figure supplement 2*), flies were grown at 25°C. Flies were then anesthetized with CO$_2$, testes removed with forceps and placed in 7 µl of PBS on (+) charged slides, the contents spilled by poking with sharp forceps, a RainX-treated coverslip placed over the testes and both snap frozen in LiN$_2$. The coverslip was then immediately removed with a razor blade and slides stored at −80°C until needed. When ready to process, slides were fixed for 20 min in 4% formaldehyde in PBT, washed three times, 5 min. each wash, in PBT. Samples were then incubated in 80% cold acetone in PBT for 10 min at −20°C and processed as per RNA-FISH for 'all larval samples' using protocol two above for detection without TSA amplification. For determination of average AAGAG(n) intensity levels, for each condition at least three testes were imaged, and at least 5 S5 spermatocytes derived from each of these testes were imaged.

## Immuno-fluorescence in adult testes without RNA-FISH

Flies were grown at 29°C and processed as above in 'RNA-FISH in adult testes' up until −80°C storage. Samples were then fixed 20 min in 4% formaldehyde in PBT, passed through an ethanol series (75–85–95%) at −20°C and dried prior to permeabilisation in 1X PBS-0.4% Triton X-100 (0.4 PBT). Samples were then blocked in 0.1PBT with 1% BSA for 1 hr at room temperature, incubated with primary antibodies overnight at 4°C and with secondary antibodies for 1 hr at room temperature (see *Table 3* for antibody information).

## Northern blotting

Non-radioactive, denaturing northern blots were essentially carried out according to Chemiluminescent Nucleic Acid Detection Module Kit (Thermofisher cat# 89880). Essentially, purified RNA was denatured for 3 min at 70°C in NorthernMax formaldehyde loading dye. Samples were then run on denaturing agarose gels with 6.9% formaldehyde in MOPS buffer. RNA was transferred to (+) charged nylon membranes in an electroblotter (FisherBiotech Semi-Dry blotting unit, FB-SDB-2020) using 200mA for 30 min. The membrane was then UVC crosslinked and prehybridized with ULTRAhyb Ultrasensitive Hybridization Buffer (Thermofisher, cat# 8669) at 68°C for 30 min. Biotinylated probes at a concentration of 30 ng/ml were then added to UltraHyb buffer, pre-hybridization

**Table 3.** Antibodies used for Immuno-Fluorescence.

| Antibody | Supplier; Cat. number | Working concentration |
|---|---|---|
| Rabbit-anti H3K9me3 | Abcam; 8898 | 1/250 |
| Mouse-anti H3K9me2 | Active Motif; 39753 | 1/250 |
| Rabbit-anti-H2AV | Lake placid AM318; 9751 | 1/100 |
| Goat anti-GFP | Rockland 121600-101-215 | 1/500 |
| Rabbit anti-H4acetyl | Millipore 06–598 | 1/200 |
| Rat anti-Mst77F | Elaine Dunleavy, PhD; NUI Galway, Ireland | 1/200 |
| Guinea pig anti-Mst35Ba/Bb (Protamine A/B) | Elaine Dunleavy, PhD; NUI Galway, Ireland | 1/200 |
| Mouse anti pan-histone | Millipore MAB 3422 | 1/200 |

solution replaced with solution containing probe and hybridized overnight at 68°C with rotation. The next day, membranes were washed and processed according to Chemiluminescent Nuclei Acid Detection kit manual. For Northern blots shown in *Figure 1—figure supplement 2*, at least three northern blots from three biological replicates were performed with similar patterns. For Northern blot shown in *Figure 1—figure supplement 6*, at least two biological replicates for each genotype were performed, with similar knockdown results.

## Identification of genomic sources of AAGAG RNA

To identify the genomic origin of AAGAG RNA, we mined *D. melanogaster* transcriptome data (modENCODE staged embryo and L3 larvae total RNA-seq reads) (*Brown et al., 2014*) for AAGAG RNA attached to mappable ends with uniquely mapped sequences and adjacent to >50 bp blocks of annotated AAGAG(n) DNA. More specifically, we first used trim_galore to filter out adaptors and low quality sequencing reads. Reads with at least three consecutive AAGAG repeats were identified and their corresponding pair-end sequences were extracted. Including only AAGAG containing reads, assemble the other end sequences into contigs using Phrap (-vector_bound 0 -forcelevel 5 - minscore 30 -minmatch 10). We then used Blast (e-value <$10^{-5}$) to identify potential genomic locations in release 6 of *D. melanogaster* genome (*Hoskins et al., 2015*) (*Table 4*). This conservative analysis revealed that the majority of AAGAG RNA originates from 2R and X heterochromatic satellites (*Table 4* and *Figure 1—figure supplement 4*). To confirm that this computational genomic analysis identified sources of AAGAG transcripts, we performed northern blotting and RNA-FISH to these and a 3R heterochromatic region. Essentially, transcript sizes using probes to these regions are similar if not identical to AAGAG RNA, and foci from these mappable regions co-localize with AAGAG RNA foci (*Figure 1—figure supplement 4, B and C–H*, respectively), demonstrating that AAGAG RNA originates from identified 2R, X and 3R heterochromatin genomic regions.

**Table 4.** Uniquely mapped RNA identified via phrap adjacent AAGAG(>10) containing blocks

| Chr | e0-2hr | e2-4hr | e4-8hr | e8-12hr | e12-14hr | e14-16hr | e16-20hr | e20-24hr |
|---|---|---|---|---|---|---|---|---|
| 2R | NA | NA | NA | NA | NA | NA | chr2R. 1825640.1825699 | NA |
| X | NA | NA | NA | NA | NA | NA | NA | chrX. 12660077.12660134 |
| X | NA | NA | NA | NA | NA | NA | chrX. 11830795.11830858 | NA |
| X | chrX. 22453019.22453120 | NA | chrX. 22453019.22453182 | NA | chrX. 22453019.22453163 | chrX. 22453019.22453177 | chr X.22453019.22453093 | chrX. 22453019.22453196 |

## Insertion of shRNA or overexpression constructs

RNAi and overexpression lines were created via small-hairpin RNA (shRNA) to AAGAG RNA driven with the UAS/GAL4 system, or in the case of control, a scrambled RNA sequence, using genomic insertion of the pValium20 vector used for the Transgenic RNAi project (TRiP) at Harvard (*Ni et al., 2011*). Importantly, the scrambled shRNA sequence contained the same percentage of A's and G's but in a random order (see *Table 5* for sequences). pValium20 constructs with shRNA or overexpression sequences (see next) were injected and screened for insertion by Rainbow Transgenic, Inc.

## Cloning of shRNA and over-expression constructs into pValium20 vector

Sense and antisense strands were annealed and ligated into digested pValium20 vector . For annealing, in a 50 μl final volume, 1.5 μl each of 100 μM stock oligos were added to 1X NEBuffer, incubated 4 min at 95℃, then slowly cooled to RT in a 1L beaker filled with 70℃ water. Samples were blunt ended with klenow using standard procedures, purified with min-elute PCR purification kit, run on agarose gel, and appropriate size bands removed and purified. Purified bands were digested with Nhe1 and EcoR1 HF enzymes and purified with min-elute PCR purification kit. For cloning, 1 μl of annealed and purified oligo pair complement was added to 30 ng of digested pValium20 vector and ligated with T4 DNA ligase (not quick ligase) at 16℃ overnight, and transformed into dh5alpha *E. coli* cells.

## Viability assay

y[1] v[1]:UAS-AAGAGshRNA:: (shRNA to AAGAG), y[1] v[1]:UAS-scramble shRNA:: (shRNA to scrambled) y[1] sc[*] v[1]; P{y[+t7.7] v[+t1.8]=VALIUM20-mCherry}attP2 (dsRNA to mCherry) males were crossed to y[1] w[*]:: P{w[+mC]=Act5 C-GAL4}17bFO1/TM6B, Tb[1] (actin-GAL4/Tubby) female virgins (see *Table 6* for fly lines). For calculation of ratios of RNAi/Tubby control prior to pupal stage (*Figure 1—figure supplement 6G*), the numbers of non-Tubby (RNAi) and Tubby pupae were scored. For each parental cross, a minimum of 11 biological replicates were completed at 25℃, and each vial included at least eight and no more than 43 pupae of any individual genotype. p-value (two tailed, type 3): **p=0.013. For calculation of death rates during different stages of development, (*Figure 1—figure supplement 6H*), we used the following: To determine L1-L2 death rates, L1 and L2 Tubby and non-Tubby (RNAi) larvae were transferred to separate vials. Those that did not survive to visible L3 were scored as dead. To determine L3 death rates, L3 from lay plates were transferred to vials and those that did not survive to pupae were scored as dead. For pupal lethality, non-eclosed pupae from L1-L2, and L3 transfers were scored as dead. L1-L2 death rate (min. n L1/L2 analyzed per parental set of three experiments = 7 L1/L2): p-values (two tailed, type 3): A/M = AAGAG to mCherry; A/S = AAGAG to Scrambled; S/M = scrambled to mCherry; A/S = 0.457, A/M = 0.404; L3 death rate (min. n per five parental sets of L3 analyzed = 7 L3): A/S = 0.125, A/M = 0.019; Pupal death rate (min. n per three parental sets of pupae analyzed = 10 pupae): A/S = 0.002, A/M = 0.992, S/M = 0.002. Of note, the high pupal death in scrambled control is perplexing considering that we could not find mRNAs that would be targeted by this hairpin. We speculate

**Table 5.** shRNA and overexpression oligos.

| Description | Sequence 5'−3' |
| --- | --- |
| shRNA to AAGAG(n) | ctagcagtGAAGAGAAGAGAAGAGAAGAGtagttatattcaagcataCTCTTCTCTTCTCTTCTCTTCgcg |
| shRNA to AAGAG(n) complement | aattcgcGAAGAGAAGAGAAGAGAAGAGtatgcttgaatataactaCTCTTCTCTTCTCTTCTCTTCactg |
| shRNA to scrambled | ctagcagtGAGAGAAAAAGGGAAAGAAGGtagttatattcaagcataCCTTCTTTCCCTTTTTCTCTCgcg |
| shRNA to scrambled complement | aattcgcGAGAGAAAAAGGGAAAGAAGGtatgcttgaatataactaCCTTCTTTCCCTTTTTCTCTCactg |
| AAGAG(37) for over-expression | ATCAAGACTGCTAGCAAGAGAAGAGAAGAGAAGAGAAGAGAAGAGAAGAGAAGAGAAGAGAAGAGAAGAGAAGAGAAGAGAAGGAAGAGAAGAGAAGAGAAGAGAAGAGAAGAGAAGAGAAGAGAAGAGAAGAGAAGAGAAGAGAAGAGAAGAGAAGAGAAGAGAAGAGAAGAGAAGAGAAGAGAAGAGAAGAGAAGAGAGAGAAGAG |
| AAGAG(37) over-expression complement | CCATTGACTGAATTCCTCTTCTCTTCTCTTCTCTTCTCTTCTCTTCTCTTCTCTTCTCTTCTCTTCTCTTCTCTTCTCTTCTCTTCTCTTCTCTTCTCTTCTCTTCTCTTCTCTTCTCTTCTCTTCTCTTCTCTTCTCTTCTCTTCTCTTCTCTTCTCTTCTCTTCTCTTCTCTT |

**Table 6.** Fly lines.

| Stock name or genotype | Obtained from: stock number | Description |
|---|---|---|
| y[1] v[1]; P{y[+t7.7]=CaryP}attP40 | Bloomington: 36304 | Background strain for insertion of pValium20 vector containing shRNA |
| y[1] v[1]; P{y[+t7.7]=CaryP}attP2 | Bloomington: 36303 | Background strain for insertion of pValium vector containing AAGAG expression construct. |
| y[1] sc[*] v[1]; P{y[+t7.7] v[+t1.8]=VALIUM20-mCherry}attP2 | Bloomington: 35785 | Control strain for RNAi. Expresses dsRNA to mCherry |
| y[1] v[1]: UAS-AAGAG shRNA:: | Rainbow Transgenic Flies, Inc | Expresses shRNA under UAS promoter targeting AAGAG(n) |
| y[1] v[1]: UAS-scramble shRNA:: | Rainbow Transgenic Flies, Inc | Expresses shRNA under UAS promoter targeting random AG containing sequences |
| y[1] w[67c23]; P{w[+mC]=dpp-GAL4.PS}6A/TM3, Ser[1] | Bloomington: 7007 | Dpp-GAL4 |
| y[1] v[1]::UAS-AAGAG(37) | Rainbow Transgenic Flies, Inc | Expresses a 187 base repeat of AAGAG RNA under a UAS promoter |
| C(1;Y)1, y[1] w[A738]: y[+]/0 and C(1)RM, y[1] v[1]/0 | Bloomington:2494 | XO (Y chromosome deficient males) |
| y[*] w[*]; P{w[+mW.hs]=GawB}NP1233/CyO, P{w[-]=UAS lacZ.UW14}UW14 | Kyoto: 103948 | Fascillin-GAL4 |
| y[*] w[*]; P{w[+mW.hs]=GawB}NP1624/CyO, P{w[-]=UAS lacZ.UW14}UW14 | Kyoto:104055 | Traffic Jam-GAL4 |
| w[*]; P{w[+mW.hs]=GawB}ptc[559.1] | Kyoto: 103948 | PTC-GAL4 |
| :: nanos-Gal4, dcr2-UAS/TM3 sb | Unknown | Nanos-GAL4 |
| w;;bamGAL4, UAS-dicer2 | Unknown | Bam-GAL4 |
| y[1] w[*]::P{w[+mC]=Act5 C-GAL4}17bFO1/TM6B, Tb[1] | Bloomington: stock 3954 | Expresses GAL4 ubiquitously under control of Act5C promoter |

that this lethality results from off-target effects on un-annotated RNA, and/or the hairpin RNA is toxic. Importantly, however, the lethal phase differed between AAGAG RNAi (L1-L3) vs scrambled RNAi (pupal) (*Figure 1—figure supplement 6H*).

## Fertility assay

Flies containing shRNA to AAGAG or scrambled control were mated to different testes GAL4 drivers (*Table 1*) at 25°C, in at least duplicate parental ($F_0$) sets. From each parental set, individual $F_1$ male progeny (minimum of 12 per parental set) were then allowed to mate with two female Oregon R virgins for 10 days at 25°C. Male flies were counted as sterile if, after 10 days, the male and at least one female were still alive and no larvae, pupae or adult $F_2$ progeny present. Female fertility was calculated as above, with one female RNAi and two Oregon R males. For Bam-GAL4-driven RNAi, female fertility was calculated as above from a minimum of three parental ($F_0$) sets using a minimum of 10 $F_1$ progeny for each. Scrambled RNAi male fertility for this cross was calculated as above from a minimum of four parental ($F_0$) sets, using a minimum of 11 $F_1$ progeny.

AAGAG Bam-GAL4-driven RNAi male fertility of 0% was calculated from >>10 ($F_0$) parental sets, hundreds of $F_1$ individual males, and at both 25°C and 29°C. For rescue experiments, triplicate parental sets were used, where one $F_1$ male (minimum 15 per parental set) was mated to three Oregon R virgin females for 10 days and fertility assayed as above.

## Morphology defects in RNAi sperm

For quantification of abnormalities in sperm DNA morphology (*Figure 3E* and *Figure 3—figure supplement 1B*), a minimum of 6 testes, each from a different male, were analyzed per genotype (see

Tables *7* and *8* below). Essentially, a projection image of the basal end of testes was made using a 40x confocal objective and all sperm DNA bundles were scored. See *Figure 3B* and *Figure 3—figure supplement 1A* for examples of sperm DNA morphology. Calculations were based on the pooled percent of a given phenotype compared to total sperm bundles per genotype.

**Table 7.** Quantification of post-canoe stage sperm DNA morphological defects in 4–7 day old testes.

| | N | Normal bundle | Lagging bundle | Kinked | Knotted | Needle eyed | Decondensed |
|---|---|---|---|---|---|---|---|
| Scrambled RNAi | 1 | 2 | 2 | 4 | 0 | 0 | 0 |
| | 2 | 9 | 7 | 6 | 0 | 0 | 0 |
| | 3 | 21 | 5 | 0 | 0 | 0 | 0 |
| | 4 | 5 | 1 | 0 | 0 | 0 | 0 |
| | 5 | 29 | 1 | 0 | 0 | 0 | 0 |
| | 6 | 6 | 2 | 0 | 0 | 0 | 0 |
| AAGAG RNAi | 1 | 0 | 8 | 2 | 0 | 0 | 2 |
| | 2 | 1 | 8 | 0 | 2 | 2 | 0 |
| | 3 | 0 | 1 | 5 | 2 | 1 | 0 |
| | 4 | 0 | 0 | 1 | 0 | 0 | 0 |
| | 5 | 0 | 1 | 2 | 1 | 1 | 0 |
| | 6 | 0 | 1 | 5 | 1 | 0 | 0 |
| | 7 | 0 | 0 | 3 | 3 | 0 | 0 |
| | 8 | 0 | 0 | 0 | 0 | 1 | 0 |
| | 9 | 1 | 2 | 4 | 1 | 1 | 0 |
| | 10 | 3 | 5 | 2 | 4 | 2 | 0 |
| | 11 | 2 | 2 | 14 | 0 | 1 | 0 |
| | 12 | 1 | 4 | 9 | 1 | 1 | 0 |
| AAGAG RNA (Rescue) | 1 | 0 | 1 | 2 | 0 | 0 | 0 |
| | 2 | 5 | 2 | 2 | 0 | 0 | 0 |
| | 3 | 7 | 1 | 1 | 0 | 0 | 0 |
| | 4 | 8 | 3 | 6 | 3 | 0 | 0 |
| | 5 | 0 | 0 | 9 | 0 | 0 | 0 |
| | 6 | 4 | 4 | 6 | 0 | 0 | 0 |
| | 7 | 8 | 4 | 3 | 0 | 0 | 0 |
| | 8 | 9 | 6 | 7 | 0 | 0 | 0 |
| | 9 | 11 | 1 | 5 | 0 | 0 | 0 |
| | 10 | 8 | 2 | 4 | 0 | 0 | 0 |
| | 11 | 3 | 0 | 2 | 0 | 0 | 0 |
| | 12 | 3 | 0 | 6 | 1 | 0 | 0 |
| | 13 | 2 | 2 | 8 | 0 | 0 | 0 |
| | 14 | 5 | 7 | 5 | 0 | 1 | 0 |
| | 15 | 5 | 0 | 3 | 0 | 0 | 0 |
| | 16 | 5 | 5 | 5 | 1 | 0 | 0 |
| | 17 | 3 | 4 | 3 | 0 | 0 | 0 |
| | 18 | 5 | 1 | 6 | 0 | 0 | 0 |
| | 19 | 3 | 11 | 5 | 1 | 0 | 0 |
| | 20 | 3 | 2 | 1 | 0 | 1 | 0 |

**Table 8.** Quantification of canoe stage DNA morphological defects, in 4–7 day old testes

|  | N | Normal canoe | Abnormal canoe |
|---|---|---|---|
| Scrambled RNAi | 1 | 2 | 0 |
|  | 2 | 7 | 1 |
|  | 3 | 6 | 1 |
|  | 4 | 3 | 5 |
|  | 5 | 9 | 1 |
|  | 6 | 5 | 1 |
| AAGAG RNAi | 1 | 1 | 0 |
|  | 2 | 0 | 1 |
|  | 3 | 0 | 1 |
|  | 4 | 1 | 1 |
|  | 5 | 0 | 0 |
|  | 6 | 0 | 1 |
|  | 7 | 0 | 0 |
|  | 8 | 2 | 4 |
|  | 9 | 1 | 2 |
|  | 10 | 0 | 2 |
|  | 11 | 1 | 2 |
|  | 12 | 3 | 6 |
|  | 13 | 2 | 7 |
|  | 14 | 1 | 5 |
| AAGAG RNA (Rescue) | 1 | 7 | 4 |
|  | 2 | 3 | 2 |
|  | 3 | 0 | 3 |
|  | 4 | 0 | 4 |
|  | 5 | 0 | 3 |
|  | 6 | 0 | 0 |
|  | 7 | 1 | 4 |
|  | 8 | 3 | 4 |
|  | 9 | 6 | 8 |
|  | 10 | 3 | 3 |
|  | 11 | 1 | 0 |
|  | 12 | 2 | 2 |
|  | 13 | 8 | 9 |
|  | 14 | 1 | 6 |
|  | 15 | 0 | 2 |
|  | 16 | 7 | 9 |
|  | 17 | 0 | 1 |
|  | 18 | 3 | 2 |
|  | 19 | 2 | 4 |
|  | 20 | 1 | 1 |

**Table 9.** qPCR oligos.

| mRNA target | Sequence 5'–3' |
| --- | --- |
| Actin-5c Forward | CAGCCAGCAGTCGTCTAATC |
| Actin-5c Reverse | ACAACCAGAGCAGCAACTTC |
| Rpl32 Forward | CGATGTTGG GCATCAGATAC |
| Rpl32 Reverse | CCCAAGATCGTGAAGAAGC |
| pip5K59B Forward | CTCCTGCTCTGCTATCGTATTC |
| pip5K59B Reverse | AGAGGAGCCATCAACATCAC |
| Peb Forward | TGGTTGGACCGCTTAACATAG |
| Peb Reverse | GCGACACCAAGAGCCATAA |
| CG33080 Forward | ATTACGATCGCGGGCTTATC |
| CG33080 Reverse | CGGTTCTAGGAGCACTGATATAAA |

## Cross to make XO males

For analysis of AAGAG RNA levels in male testes without a Y-chromosome, y[1]w[1] males were mated to C(1)RM, y[1] v[1]/0 females (Bloomington stock # 2494) at 25°C and 0–6 hr testes from $F_1$ males imaged.

## qPCR conditions

RNA was extracted and cDNA made by established methods. For qPCR, 10 µl 2X Absolute Blue qPCR SYBR low Rox mix (Thermofisher, cat. AB4318) was added, forward and reverse oligos each to 0.15 µM, 0.5 µl cDNA, and water to 20 µl. qPCR conditions were as follows: 95°C, 15 min; 40cycles (95°C 15 s, 58°C 30 s., 72°C 30 s); 72°C 30 s performed on AB 7500 Fast Real Time PCR System. Performed in biological triplicates. See *Table 9* for qPCR oligos.

## Acknowledgements

We thank Karen Miga and Miten Jain for RNA-seq analysis efforts, Shelby Wilson for technical guidance with spermatogenesis analysis, and Aniek Janssen for editing. We also thank Caitriona Collins for making the transition protamine and protamine antibodies.

## Additional information

### Funding

| Funder | Grant reference number | Author |
| --- | --- | --- |
| National Institutes of Health | RO1 GM117420 | Gary H Karpen |
| National Science Foundation | Graduate Research Fellowship | Wilbur Kyle Mills |
| Science Foundation Ireland–Health Research Board–Wellcome Trust (SFI-HRB-WT) | 00105/Z/12/Z | Elaine M Dunleavy |
| National Institutes of Health | K99 GM121868 | Yuh Chwen G Lee |
| Science Foundation Ireland – President of Ireland Young Research Award (SFI-PIYRA) | 13/YI/2187 | Antje M Kochendoerfer Elaine M Dunleavy |

The funders had no role in study design, data collection and interpretation, or the decision to submit the work for publication.

## Author contributions
Wilbur Kyle Mills, Conceptualization, Data curation, Formal analysis, Funding acquisition, Investigation, Methodology, Project administration, Supervision, Validation, Visualization, Writing—original draft, Writing—review and editing; Yuh Chwen G Lee, Data curation, Formal analysis, Investigation, Visualization, Methodology, Writing—original draft, Writing—review and editing; Antje M Kochendoerfer, Data curation, Formal analysis, Investigation, Visualization, Methodology, Writing—review and editing; Elaine M Dunleavy, Resources, Data curation, Formal analysis, Supervision, Investigation, Visualization, Methodology, Writing—original draft, Project administration, Writing—review and editing; Gary H Karpen, Conceptualization, Resources, Formal analysis, Supervision, Funding acquisition, Investigation, Visualization, Methodology, Writing—original draft, Project administration, Writing—review and editing

## Author ORCIDs
Wilbur Kyle Mills (iD) https://orcid.org/0000-0002-4842-4190
Yuh Chwen G Lee (iD) https://orcid.org/0000-0002-0081-7892
Antje M Kochendoerfer (iD) https://orcid.org/0000-0001-5862-667X
Elaine M Dunleavy (iD) https://orcid.org/0000-0001-6885-0421
Gary H Karpen (iD) https://orcid.org/0000-0003-1534-0385

## Decision letter and Author response
Decision letter https://doi.org/10.7554/eLife.48940.sa1
Author response https://doi.org/10.7554/eLife.48940.sa2

# Additional files

## Supplementary files
• Transparent reporting form

## Data availability
All generated data are included within the article.

The following datasets were generated:

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
