## [Decision Letter]

**Acceptance summary:**

Transcription from repetitive DNA can nucleate heterochromatin formation, a structural role using RNA to decorate chromatin. Mills et al. show a surprising and focused requirement of a simple repeat RNA in development. This work shows that the AAGAG tandem repeat long noncoding RNA localizes in nuclear foci, primes *Drosophila* male germ line development, and is essential for male fertility. Although the mechanism of action is still unclear, these results highlight novel roles of nuclear noncoding RNA in chromatin organization and provided insights about in vivo functions that depends on lncRNA.

**Decision letter after peer review:**

Thank you for submitting your article "RNA from a simple-tandem repeat is required for male fertility and histone-protamine exchange in *Drosophila melanogaster*" for consideration by *eLife*. Your article has been reviewed by three peer reviewers, one of whom is a member of our Board of Reviewing Editors, and the evaluation has been overseen by James Manley as the Senior Editor. The following individual involved in review of your submission has agreed to reveal their identity: Yukiko M Yamashita (Reviewer #3).

The reviewers have discussed the reviews with one another and the Reviewing Editor has drafted this decision to help you prepare a revised submission.

Essential revisions:

The role of satellite RNA in spermatogenesis as presented in this study is extremely interesting, but there are major concerns, which must be addressed prior to publication.

1) The authors use one short hairpin RNA (shRNA) construct to knockdown AAGAG RNAs and observe several phenotypes. To their credit, the authors show that the knockdown phenotype on fertility can be partially reversed by ectopic overexpression of the RNA. However, it was surprising that shRNA against mCherry also greatly reduces the AAGAG RNA (Figure 1—figure supplement 6). This raises concerns about the specificity of the experiment. Moreover, if the AAGAG RNA can be reduced but without phenotype in the mCherry shRNA experiments, then the concern is that the observed phenotypes are due to an off-target or non-specific effect of the particular shRNA. Independent RNAi lines or orthogonal methods to deplete or block AAGAG RNA are necessary to confirm the phenotype. Given that this study focuses on spermatogenesis, RNAi validation (AAGAG RNA-FISH) on AAGAG RNAi testes must also be shown.

2) Whereas AAGAG RNAi phenotype is consistent with histone-protamine transition defects, many other upstream causes could lead to histone-protamine transition defects. Accordingly, the observed phenotype does not necessarily pinpoint AAGAG's function in histone-protamine exchange, and thus AAGAG RNA's function remains undefined. But the manuscript is written in a way to strongly indicate AAGAG is directly involved in a 'chromatin based' mechanism that promotes histone-protamine exchange. We suggest that the authors be more cautious in writing (we are not asking to provide 'mechanism' for AAGAG function in histone-protamine transition).

3) The Northerns need clarification. The authors show that a 1.5-kb maternal product is loaded, and then RNA levels increase throughout development. There are several technical questions (e.g. the blots seem to cut off signal above 1.5-kb, how much of this RNA is coming from the satellite, where is the 1.5-kb signal coming from?). Regarding strands and satellite-derived RNA, it's not really clear what (+) and (–) mean. While it is ok to arbitrarily assign (+) and (–) with respect to the sequence 'AAGAG', it would be surprising if the orientation within an AAGAG satellite block didn't occasionally flip. For example, a structural rearrangement of AAGAG tandem repeats could generate both types of transcripts (AAGAG and CUCUU) off of one strand. The presence of CUCUU in the spermatogenesis doesn't necessarily mean that the same repeats are transcribed off of both strands.

---

## [Author Response]

Essential revisions:The role of satellite RNA in spermatogenesis as presented in this study is extremely interesting, but there are major concerns, which must be addressed prior to publication.1) The authors use one short hairpin RNA (shRNA) construct to knockdown AAGAG RNAs and observe several phenotypes. To their credit, the authors show that the knockdown phenotype on fertility can be partially reversed by ectopic overexpression of the RNA. However, it was surprising that shRNA against mCherry also greatly reduces the AAGAG RNA (Figure 1—figure supplement 6). This raises concerns about the specificity of the experiment. Moreover, if the AAGAG RNA can be reduced but without phenotype in the mCherry shRNA experiments, then the concern is that the observed phenotypes are due to an off-target or non-specific effect of the particular shRNA. Independent RNAi lines or orthogonal methods to deplete or block AAGAG RNA are necessary to confirm the phenotype.

We are sorry the way we presented this data caused confusion about the specificity of RNAi. Although the AAGAG RNA signal in mCherry RNAi control lane appears weak, as stated in the legend “The AAGAG RNAi L3 AAGAG RNA top band signal is approximately 86% and 75% reduced compared to either scrambled or mCherry controls, respectively, when normalized to the actin-5c loading control.” Thus, quantitation with proper normalization shows that AAGAG RNA is greatly reduced after AAGAG RNAi compared to either control (though there is a bit less depletion for mCherry vs. Scrambled). Yes, the Northern loading is not optimal, but producing a new Northern is greatly hampered by the departure of the lead and second authors from the lab.

More importantly, there are multiple lines of evidence supporting the specificity of the RNAi with respect to the observed phenotypes;

1) The Northern results are validated by the AAGAG RNA-FISH (Figure 1—figure supplements 6A-D, in which we see abolishment of AAGAG RNA foci in L3 brain lobes with AAGAG RNAi but not in scrambled or mCherry controls. Additionally, we calculated the number of AAGAG RNA foci per nuclei with the sections we analyzed and find that when analyzing >33 nuclei per section we see roughly 20% of mCherry nuclei per section have AAGAG RNA foci while 14% of scrambled RNAi have AAGAG RNA foci. Note that we analyzed the outermost sections of the brain for each genotype to eliminate artifacts from poor probe/antibody entry into deep tissue, and these numbers are only from confocal sections, and not from projections. This indicates that mCherry RNAi does not drastically affect AAGAG RNA foci compared to scrambled control, and both are distinct from the effects of AAGAG RNAi.

2) The highly penetrant phenotypes are only observed after AAGAG RNAi, and not with either control.

3) The phenotypes are partially (but significantly) rescued by addition of AAGAG RNA.

Thus, we believe that concerns about the phenotypes and off-target effects are not warranted. Note that we did try ‘gapmers’, but they did not knockdown AAGAG RNA and exhibited high toxicity after injection into syncytial embryos. We are not aware of other methods, including all current versions of CRISPR-Cas9, etc., that would, without significant development and optimization in flies, be useful for specific knockdown of this type of RNA without affecting underlying DNA or without producing unexpected phenotypes.

Given that this study focuses on spermatogenesis, RNAi validation (AAGAG RNA-FISH) on AAGAG RNAi testes must also be shown.

RNAi validation of AAGAG RNAi knockdown of AAGAG in testes is shown in Figure 2—figure supplement 3.

2) Whereas AAGAG RNAi phenotype is consistent with histone-protamine transition defects, many other upstream causes could lead to histone-protamine transition defects. Accordingly, the observed phenotype does not necessarily pinpoint AAGAG's function in histone-protamine exchange, and thus AAGAG RNA's function remains undefined. But the manuscript is written in a way to strongly indicate AAGAG is directly involved in a 'chromatin based' mechanism that promotes histone-protamine exchange. We suggest that the authors be more cautious in writing (we are not asking to provide 'mechanism' for AAGAG function in histone-protamine transition).

We agree that it is unclear if AAGAG functions directly in histone-protamine exchange and did not intend to strongly convey that conclusion. This is a striking phenotype, but there are many defects observed during sperm maturation. Further, direct effects (in a temporal sense) are unlikely given that all phenotypes are only visible when the RNA is *not* visible in wild-type, well after the primary spermatocyte stage. We have tried to alter the text to state that AAGAG RNA expression in primary spermatocytes is necessary for normal spermatogenesis, loss of AAGAG RNA causes the phenotypes, and direct vs. indirect effects cannot be distinguished.

That said, we have changed the title to emphasize effects on sperm maturation and fertility, rather than histone-protamine exchange.

We also included more information showing that AAGAG(37) RNA expression in primary spermatocytes partially rescues the sperm defects seen with AAGAG RNA depletion in primary spermatocytes. Specifically, we added new data to Figure 3E, confirming that rescue with AAGAG(37) RNA expression in primary spermatocytes in AAGAG RNAi background partially rescues individualization defects (only reported% fertile in previous version).

Additionally, we added a new figure, (Figure 3—figure supplement 1) showing that histones retained in AAGAG RNAi testes up to late canoe stage, but not in the control. This is an interesting finding considering that histones are generally removed just prior to late stage canoe and lends more support that AAGAG RNA affects histone:protamine transition (of course without addressing direct vs. indirect mechanism).

3) The Northerns need clarification. The authors show that a 1.5-kb maternal product is loaded, and then RNA levels increase throughout development.

We do not claim that RNA levels increase throughout development, although it is likely given the increased signal in the Northerns throughout development. Similar levels of RNA were loaded per well (according to Qubit RNA high sensitivity measurements) and similar exposure times were used, but it is impossible to quantify levels between 0-24hr, larval vs. maternal and 2-4hr due to the absence of loading controls and separate northerns for later individual stages. We do not want to say anything at this time about the levels of RNA during development or in different tissues, so if any text suggests changes remains please point them out so we can remove them.

There are several technical questions (e.g. the blots seem to cut off signal above 1.5-kb, how much of this RNA is coming from the satellite, where is the 1.5-kb signal coming from?).

Figure 1—figure supplement 2 Northern blots now have a larger cutoff and show that the 1,500 nt band is the largest band present. We speculate that this signal is coming from the AAGAG(n) regions adjacent chr3R:3,169,758-3,169,820, chr2R:1,826,691-1,826,740 and/or chrX:22,453,019-22,453,076 regions, as northern blotting to each of these regions produces a 1,500 nt band (Figure 1—figure supplement 4B). It is unclear why they all present as a 1,500 nt band, although it is possible they originate in different sizes and are processed to 1,500 nt or smaller.

Regarding strands and satellite-derived RNA, it's not really clear what (+) and (–) mean. While it is ok to arbitrarily assign (+) and (–) with respect to the sequence 'AAGAG', it would be surprising if the orientation within an AAGAG satellite block didn't occasionally flip. For example, a structural rearrangement of AAGAG tandem repeats could generate both types of transcripts (AAGAG and CUCUU) off of one strand.

We agree that there could be ‘flipping’ and other structural changes, and we struggled with how to present this complex experiment and data. We intend to use the (+) and (–) only in reference to the probes and annotated DNA sequence, not putative adjacent sequences that may have ‘flipped’ or changed. Figure 1—figure supplement 4A has been changed to provide more clarification and indicate where the probes will bind when designated (+) and (-). Text has also been changed to indicate that orientation of an AAGAG satellite block may occasionally flip (though note it is striking that in embryos we cannot find the complementary RNA foci…if flipping occurred we’d expect to see some).

The presence of CUCUU in the spermatogenesis doesn't necessarily mean that the same repeats are transcribed off of both strands.

We agree, and did not intend to state they are transcribed off both strands. Again, we want to know if specific text remains that suggests this is the case.